

# Advancing land surface model development with satellite-based Earth observations

Rene Orth[1,2], Emanuel Dutra[1], Isabel F. Trigo[3,4], and Gianpaolo Balsamo[1]

[1] European Centre for Medium-Range Weather Forecasts, Shinfield Park, Reading RG2 9AX, UK
5 [2] Institute for Atmospheric and Climate Science, ETH Zürich, Universitätstrasse 16, CH-8092 Zürich, Switzerland
[3] Instituto Português do Mar e da Atmosfera, 1749-077 Lisboa, Portugal
[4] Instituto Dom Luiz (IDL), 1749-016 Lisboa, Portugal

*Correspondence to:* R. Orth (rene.orth@env.ethz.ch)

10 **Abstract.** The land surface forms an essential part of the climate system. It interacts with the atmosphere through the exchange of water and energy and hence influences weather and climate, as well as their predictability. Correspondingly, the land surface model (LSM) is an essential part of any weather forecasting system. LSMs rely on partly poorly constrained parameters, due to sparse land surface observations. With the use of newly available land surface temperature observations, we show in this study that novel satellite-derived 15 datasets help to improve LSM configuration, and hence can contribute to improved weather predictability.

We use the *Hydrology Tiled ECMWF Scheme of Surface Exchanges over Land* (HTESSEL) and validate it comprehensively against an array of Earth observation reference datasets, including the new land surface temperature product. This reveals satisfactory model performance in terms of hydrology, but poor performance in terms of land surface temperature. This is due to inconsistencies of process representations in the model as 20 identified from an analysis of perturbed parameter simulations. We show that HTESSEL can be more robustly calibrated with multiple instead of single reference datasets as this mitigates the impact of the structural inconsistencies. Finally, performing coupled global weather forecasts we find that a more robust calibration of HTESSEL also contributes to improved weather forecast skills.

In summary, new satellite-based Earth observations are shown to enhance the multi-dataset calibration of LSMs, 25 thereby improving the representation of insufficiently captured processes, advancing weather predictability and understanding of climate system feedbacks.

## 1. Introduction

The land surface forms an essential part of the climate system. It interacts with the atmosphere through the exchange of water and energy and hence influences weather and climate (Seneviratne et al. 2010). Soils, 30 vegetation and water bodies store large amounts of energy and moisture. Through this storage and control capacity, the land surface can accumulate and maintain anomalies induced by the atmospheric forcing (Orth et al. 2013). These persistence characteristics and the associated predictability make the land surface an important potential contributor of weather and climate forecast skill (Orth and Seneviratne 2014, Orth et al. 2016). Furthermore, the land surface can play an important role during extreme events (Mueller and Seneviratne 2013, 35 Miralles et al. 2014, Hauser et al. 2016). For instance dry soils can contribute to the intensification of heat waves but buffer floods, whereas wet soils can mitigate hot extremes but enhance the risk for flood events.

However, state-of-the-art land surface models have difficulties to correctly capture land surface dynamics and the related coupling with the atmosphere (Beven and Binley 1992, Beven 2001, Wang et al. 2014, Trigo et al. 2015) and show margins for improvement when compared to simple well-tuned models (Best et al. 2015, 40 Haughton et al. 2016). This is hampering a full exploitation of related predictability, and the accurate representation of extreme events.



The shortcomings of the models are partly related with sparse observations and the spatial heterogeneity of soils and vegetation. Until recently, available observations were not sufficient to satisfactory constrain complex land surface models which include relevant physical processes required to represent the land-atmosphere coupling. This lead to the paradox situation that these complex models could not outperform simple conceptual models

with a very simplified representation of processes, as these can be more accurately calibrated with the few available observations (Orth and Seneviratne 2015, Best et al. 2015).

This might change in the coming years thanks to new satellite-derived datasets which become increasingly available. Related products are already available for essential variables such as surface soil moisture (Liu et et al. 2011, Liu et al. 2012, Wagner et al. 2012) or terrestrial water storage (Swenson and Wahr 2006, Landerer and

Swenson 2012). Also information on the heterogeneity of the land surface has strongly improved thanks to satellite-based observations (e.g. global land cover facility, http://glcf.umd.edu, and harmonized world soil database, http://webarchive.iiasa.ac.at/Research/LUC/External-World-soil-database/HTML/ [accessed on 11 September 2016]). The unprecedented spatial and temporal coverage of these data offer the potential to enhance the calibration/optimization of unconstrained parameters in   land surface models taking into account the

variability in soil and vegetation types.

In this study we employ satellite-derived observations of land surface temperatures (LST) which has a high information content on the surface turbulent flux partitioning and on the global surface properties (Mildrexler et al. 2011). Surface temperatures are inferred from emitted infrared radiation a high temporal frequency such that even the diurnal cycle can be observed (Trigo et al. 2011). While the above-mentioned products help to

constrain the land water balance, LST products provide complementary information on the land energy balance. Consequently the LST data is expected to bring further constraints to the surface water/energy budgets and improve the land-atmosphere coupling in land surface models. The product considered in this study is based on data from the geostationary Meteosat Second Generation satellite and provides LST information at high temporal and spatial resolution for Europe and Africa. Especially for the latter region, such satellite-based

datasets are essential as ground observations are particularly sparse.

Previous studies used LST data from particular days or particular locations to evaluate land surface models (e.g. Wang et al. 2014, Trigo et al. 2015). It is the first objective of this study to comprehensively assess model performance at large spatial scales and with multi-year LST data. Our second objective is to use an increasing number of Earth observation datasets in addition to the LST data to demonstrate that land surface model

performance benefits from a comprehensive calibration against a wide range of observational datasets. While they all include characteristic uncertainties and shortcomings, their joint use could help to better constrain land surface models. Furthermore, by assessing land surface model output with all the employed datasets we can better understand the functioning of the model, identify inconsistencies, and insufficiently represented processes.

Finally, we also investigate the role of the land surface model calibration for the skill of related coupled weather forecasts. This way, we test to which extent an improved representation of land surface processes can propagate into the (modelled) climate system to yield improved predictions.





## 2. Methodology

In this study we follow the methodology proposed by Orth et al. 2016 (hereafter referred to as O16) regarding the modelling environments and analysis. Sections 2.1-2.4 provide an overview of the model simulations and their analysis, while full details can be found in O16. O16 used these simulations to analyze the sensitivity of the

performance of the land surface model and of the weather forecasting system with respect to particular land surface model parameters. In contrast, we will analyze to which extent the simulations capture observed LST and its dynamics, and show that the use of LST data alongside further reference datasets enables a comprehensive and robust land surface model calibration which is also beneficial for weather forecast skills.

### 2.1 Model description

#### 2.1.1 Land surface model HTESSEL

The ECMWF's land surface model Hydrology Tiled ECMWF Scheme for Surface Exchanges over Land (HTESSEL, Balsamo et al. 2011) is an integral component of the ECMWF Integrated Forecasts System (IFS), that is used in the different forecast and data assimilation systems, ranging from deterministic 10-day forecasts to the ensemble seasonal forecasts. The surface model is responsible for providing the atmospheric boundary

conditions (heat, moisture and momentum) by simulating the surface water and energy budgets and the temporal evolution of the underlying soil (temperature and moisture), snowpack and vegetation interception.

The surface energy budget is computed in each grid-box independently for different tiles representing different land cover types (e.g. bare ground, high/low vegetation). The surface energy balance is coupled to the underlying soil (or snow) via the skin conductivity, which is currently a single parameter depending on land

cover. This is a simplified approach to represent very complex processes such as within-canopy energy exchanges, while it is crucial for the LST computation.

#### 2.1.2 ECMWF ensemble prediction system

The ECMWF ensemble prediction system (Vitart et al. 2008, 2014) is used daily for global forecasts up to the monthly range and it allows to characterise the uncertainty in the meteorological forecast expressed by the

spread of the ensemble members (51-forecast realisations in the operational configuration). The spread of the ensemble varies with the difficulty of predicting a given meteorological event, due to the complex evolution of the atmospheric flow and the local climate and seasonal conditions, and it is a highly valuable information on the likelihood of the forecast being accurate. In this study we use 15-member ensemble forecasts.

### 2.2 Model simulations

Our main objectives are to study the performance of HTESSEL against multi-year LST data covering two continents, and to analyze the benefits of calibrating HTESSEL against multiple reference datasets, including LST data. For this purpose we employ two sets of simulations with perturbed model parameters. The corresponding uncoupled HTESSEL simulations, and the coupled forecasts with the ensemble prediction system (that includes HTESSEL) are listed in Table 1 and described in this section.





### 2.2.1 Uncoupled HTESSEL simulations

The use of HTESSEL in an uncoupled, stand-alone setting is computationally inexpensive and allows to perform long-term simulations across the entire European continent. We analyze 50 simulations of HTESSEL with default and perturbed parameters (see Section 2.2.3). The simulations are forced with observed meteorological

information as those used in the computation of the ERA-Interim/Land dataset (Balsamo et al. 2015), The simulations are computed for the 1983-2014 period. The first year was used to spin-up the model, which therefore is not considered in the analysis.

### 2.2.2 Coupled forecasts

Coupled forecasts with the ECMWF's ensemble prediction system were computed to assess the response of

weather forecast skills to different (parameter) configuration of the land surface model. We employ 11 sets of global forecasts with default and perturbed land surface model configurations. The analysis of the forecasts focuses on the European domain used for the uncoupled HTESSEL simulations, and on northern hemispheric summer. This allows us to study impacts of the land-atmosphere coupling in Europe as this is strongest at that time, and to exclude confounding effects from snow and ice on our analysis.

Correspondingly, the forecasts are initialized on eight start dates (1 May, 15 May, 1 June, 15 June, 1 July, 15 July, 1 August, and 15 August) during 2001-2010 and computed until 45 days lead time. Each forecast constitutes an ensemble of 15 members which enable us to perform deterministic and probabilistic skill evaluations. Note that as the forecast sets differ with respect to the land surface model configuration, also the initial land conditions for the forecasts may be different. They are taken from the uncoupled HTESSEL

simulations with the respective configuration. Consequently, the forecast skill is not only impacted by the altered HTESSEL configurations during the forecasting period, but also by correspondingly different initial land conditions. All further required initial conditions are taken from the ERA-Interim dataset (Dee et al. 2011), and from the ECMWF ocean reanalysis (Balmaseda et al. 2013).

### 2.2.3 Parameter perturbations

O16 perturbed a set of six poorly constrained parameters which are deemed important for the performance of the HTESSEL model. They are listed in Table 2.

All selected parameters are perturbed at once (Saltelli et al. 2008). For this purpose, multiplicative factors between 0.25-4 (0.5-2 in the case of the soil depth) were applied to the default values of each of the chosen parameters. The factors were determined with a quasi-random sampling approach (Sobol 1967) which allows to

efficiently sample the entire parameter space without introducing correlations between the perturbations of the considered parameters. This way, a large sample of perturbed parameter sets was generated by O16, of which 50 parameter sets were chosen to limit the computation effort for the uncoupled HTESSEL simulations covering Europe. Out of these 50 parameter sets, 25 were chosen randomly while ensuring that the resulting multiplication factors applied to particular parameters are not correlated. The remaining 25 parameter sets were

selected from corresponding HTESSEL simulations that agreed best with a suite of Earth observations at 6 locations across Europe. They include the default configuration of the model.





As coupled global forecasts are computationally demanding, they were only computed for a subset of 11 out of the 50 sets of perturbed parameters. This subset includes the default configuration, 5 configurations of the randomly chosen parameter sets, and 5 configurations of the best-performing parameter sets. For the selection of 5 (out of 25, or 24 in the case of the best-performing parameter sets as the default parameter set is already

considered for computing the forecasts), all possible sets of 5 configurations were tested to choose the configurations with the lowest correlations between the multiplicative factors of the particular parameters.

## 2.3 Performance measures

The uncoupled HTESSEL simulations and the coupled forecasts are validated against a range of reference datasets (see Section 3.2), using multiple measures of agreement introduced in this section. These different

measures allow to make more efficient use of the information contained in the reference data (Vrugt et al. 2003). In particular, we consider:

- Anomaly correlation:

  We subtract the mean seasonal cycle at each grid cell in both the model output and the reference dataset and correlate the resulting anomalies. The mean seasonal cycle is determined from the entire

considered time series at each grid cell.

- Bias:

  The bias is derived by subtracting the mean of the model output from the mean of the reference dataset at each grid cell.

While these measures are used to evaluate the uncoupled HTESSEL simulations and the coupled forecasts, we

use another measure for the coupled forecasts only:

- Reliability:

  The reliability measures the ability of ensemble forecasts to accurately capture the occurrence probability of an event. We consider four events which comprise temperature and precipitation anomalies in the lower and upper tercile, respectively. For the assessment of the reliability, all forecasts

from grid cells in a particular region are grouped with respect to the forecasted occurrence probability of a particular event. Then the observed frequency of the considered event across all forecast dates in the group is computed and compared with the forecasted occurrence probability. The resulting relationship between all groups of forecasted probabilities and the respective average observed frequencies (reliability diagram, see e.g. Weisheimer and Palmer 2014) can be assessed through a slope

of a linear least-squares regression fit (see O16 for details).





All forecast performance measures are computed for particular regions and lead times. In this context we consider the Northern, Central and Southern European regions (as introduced in Seneviratne et al. 2012), and the forecasts are averaged and evaluated for lead times between 1-15 days, 16-30 days, and 31-45 days. Forecast performances for the entire European domain in terms of anomaly correlation, bias, and reliability, respectively, are then determined by (i) ranking the forecasts obtained with the 11 HTESSEL parameter sets in each of the 3 subregions, and then (ii) ranking the sum of the resulting three ranks for each skill measure. This means the HTESSEL parameter set performing best across Europe in terms of a particular skill metric (e.g. temperature bias) must not necessarily be the best in all considered subregions, but has the lowest sum of the ranks from all subregion rankings.

In line with our forecasts that are initialized and computed during late spring and summer, the evaluation of the uncoupled HTESSEL simulations focuses on May-October to exclude impacts of ice and snow on the quality of the reference datasets and on the strength of the land-atmosphere coupling.

### 2.4 Computation of parameter sensitivities

We assess the performance of the uncoupled HTESSEL model against LST data and further reference datasets, and analyse its sensitivity to variations in particular parameters. In this context all combinations of performance metrics (considered measures of agreement with all employed reference datasets) and perturbed parameters are considered. Sensitivities are computed from the relationship between model performance and underlying multiplicative factors applied to the considered model parameter. A smoothing function (cubic spline function) is fitted to capture this model performance-multiplicative factor relationship (see Figure S1 in O16 for illustration). The sensitivity is then expressed as the fraction of performance variability captured by the smoothing function, i.e. it is calculated by dividing the performance variability captured with the smoothing by the performance variability computed across all involved multiplicative factors.

### 2.5 Processing of LST data

LST data are new satellite-derived Earth observations which help to better constrain the land energy balance that was so far only captured by the evapotranspiration reference data.

The LST dataset used in this study is available at very high spatial (geostationary projection, 3-km at the sub-satellite point) and temporal resolution (15 minutes) across the Meteosat disk (Freitas et al. 2010, Trigo et al. 2011). Here we use hourly fields re-projected onto a regular 0.05°x0.05° grid covering Europe and Africa, which were subsequently processed to obtain mean daily LSTs and the daily LST range at the resolution of the HTESSEL simulations (0.5°x0.5°). We refer to the daily LST range as the difference between the maximum and minimum hourly value of a given day at a particular location. In addition, also the modelled LST data is filtered to exclude (modelled) cloudy days from the comparison. For the data processing we follow several steps:

1. If any 0.05°x0.05° grid cell has more than 2 missing values (i.e. less than 22 hourly values) on a particular day, all data of that day is disregarded. This ensures that any daily values we compute are based on a representative set of at least 22 hourly observations.

2. Any 0.5°x0.5° grid cell is composed of 100 0.05°x0.05° grid cells. If at least 80 of these contain observations, we compute an hourly average across the available (80-100) grid cells.

3. From these hourly averages the daily mean LST and daily LST range of the particular 0.5°x0.5° grid cell is computed.





4. The modelled LST data is filtered with respect to the concurrent simulated cloud cover. HTESSEL outputs cloud cover at each grid cell for 3-hourly periods. LST data from a particular grid cell and day are considered if total cloud cover is below 10% in every 3h-period of that day.

Note that these filtering steps are rather strict to guarantee the best possible comparison with the simulations.
Some of the filtering steps might be relaxed for other applications, but a detailed evaluation of this filtering is beyond the scope of this study.

## 3. Data

### 3.1 Forcing data

As we are aiming to compare the uncoupled HTESSEL simulations against observation-based reference
datasets, we use observed meteorological forcing to compute all simulations. For this purpose we employ the WFDEI dataset (Weedon et al. 2014), which is based on bias-corrected ERA-Interim data.

### 3.2 Validation data

#### 3.2.1 Validation of uncoupled HTESSEL simulations

In addition to comprehensively evaluating the LST performance of HTESSEL, it is a main objective of this
study to analyze and illustrate the value of using an array of Earth observation datasets instead of single datasets to calibrate a land surface model. For this purpose we consider several reference datasets:

- Soil moisture:

  We use data from 11 stations across Europe. Stations are located in Finland (4), Switzerland (5), and Italy (2), and provide therefore data from all relevant European climate regimes. Data is available from
different soil depths, and during different time periods, which both vary with respect to the station (see Table S1 in Orth and Seneviratne 2015 for details). For every station, however, there are at least 4 years of data. Aggregating the data from the different depths, we derive a weighted average (with respect to observed depths) to represent soil moisture within the top meter of the soil. The same is done with the HTESSEL data, using the three uppermost soil layers.

- Total terrestrial water storage:

  This quantity is derived from satellite measurements of temporal variations in the Earth's gravity field. The resulting GRACE dataset (Swenson and Wahr 2006, Landerer and Swenson 2012) provides gridded quasi-monthly water storage anomalies, and spans from 2003-2012. We use the release of the Center for Space Research of the University of Texas at Austin. Note the relatively low spatial
resolution of about 2°x2° in Europe. These observations are compared with a weighted average (with respect to soil depth) of HTESSEL soil moisture from all model layers.

- Evapotranspiration (ET):

  Gridded monthly ET data is used from the LandFlux-EVAL dataset (Mueller et al. 2013). This dataset is a blend of diagnostic and modelled datasets. Whereas the diagnostic datasets are based on (point-
scale and satellite) observations, the modelled datasets are obtained by forcing land surface models with observed meteorological forcing. The dataset covers the period 1989-2005 and is provided at a spatial resolution of 1°x1°.



- Streamflow:

   We employ daily streamflow data from over 400 near-natural, small (~10-100 km$^2$) catchments distributed across Europe from Stahl et al. 2010. The dataset spans through 1984-2007. These observations are compared with HTESSEL streamflow data from the respective grid cell within which

(most of) a particular catchment is located.

- Land surface temperature:

   We use land surface temperature data generated by the Satellite Application Facility on Land Surface Analysis (LSA SAF, Trigo et al. 2011, Freitas et al. 2010), which is based on observations of the Spinning Enhanced Visible and InfraRed Imager (SEVIRI) onboard the Meteosat Second Generation

satellite. The gridded LST data is available for 2007-2014. We compare these data with daily skin temperature data from HTESSEL.

All these reference datasets are complementary in terms of spatial coverage and temporal availability. For example, whereas the soil moisture stations represent particular locations, the GRACE and LST data fully cover the European continent (except for cloudy regions in the latter case), but in lower spatial resolution. And while

the ET data help to constrain HTESSEL's energy balance during early years (1989-2005), the LST data cover the recent years (2007-2014).

Note that for the in-situ soil moisture and GRACE we only consider anomaly correlations to compute the agreement between the reference data and the HTESSEL output. For ET, streamflow, mean daily LST, and daily LST range we additionally consider the bias. This results in a total of 10 validation metrics which we use in our

analysis.

### 3.2.2 Validation of coupled forecasts

We determine the skill of the coupled forecasts against gridded temperature and precipitation observations from the E-OBS dataset version 12 (Haylock et al. 2008). It is based on corresponding station observation from across Europe. See Section 2.3 for the employed measures of agreement between forecasted and observed data.

## 4. Results

### 4.1 Skin temperature performance of HTESSEL

We perform the most comprehensive large-scale evaluation of LSTs from a land surface model performed so far, covering 8 years and 2 continents. The LST performance of HTESSEL using its default configuration is displayed in Figure 1. There are significant biases in mean skin temperature (>5°C in the Arabian Peninsula),

and even more in the daily LST range (up to 10°C in southern Europe and southern Africa). We find strong spatial differences in terms of the performance of the temporal LST dynamics in HTESSEL with lowest correlations in low latitudes. Interestingly, the performance in terms of biases and dynamics do not correspond, we find regions with low biases but low correlations (e.g. Sahel), or regions with strong biases but good representation of observed daily dynamics (e.g. southern Europe).





We furthermore perform this evaluation for land cover classes; for this purpose we only consider grid cells where the respective land cover accounts for more than 80% of the grid cell area. Note that consequently not all areas are included in this analysis as some regions are characterised by mixed vegetation (e.g. Europe). In the lower part of Figure 1 we find that HTESSEL's performance in simulating daily skin temperature range depends
on land cover, with better performance over less vegetated areas. In contrast, this is not the case for mean skin temperature performance. The area denoted with the dashed rectangles is the European region on which the rest of this study focuses as multiple Earth observations are available there.

In Figure 2, we analyse the sensitivity of HTESSEL's performance with respect to perturbations in selected, poorly constrained model parameters (x-axis). In this context, HTESSEL's performance is determined against
several reference datasets (y-axis). An initial analysis employing some reference datasets has been done in O16, and the corresponding results are shown in the upper part of Figure 2. We extend this analysis using LST data as displayed in the lower part of the figure. All parameters influence model performance in some respect, except for runoff depth and maximum interception. While the HTESSEL performance in terms of hydrological datasets (upper part) is sensitive mostly to stomatal conductivity, its skin temperature performance (lower part) is
especially sensitive to the minimum stomatal conductivity parameter. The performance in terms of both groups sensitive to shape of soil moisture stress function. An important implication of this is that skin temperature performance can not be improved without impacting the hydrological performance of the model. As in Figure 1, we find  an apparent influence of land cover on skin temperature performance, however, with similar parameter sensitivities across the different land covers. This suggests that a re-calibration of the model could improve skin
temperature independent of land cover.

Adding to the sensitivities determined over Europe, Figure S1 shows the sensitivity of the LST performance of HTESSEL determined over the entire domain displayed in Figure 1. The results are similar. Outside Europe we can also analyse skin temperature performance over bare soils and find similar sensitivities as for the other considered land covers. Generally, the spatially similar sensitivities support the representativeness of European
skin temperature results and suggest that improvements of European LSTs would also translate into African LSTs which correspond better with the satellite observations.

### 4.2  Added value of calibrating HTESSEL against multiple reference datasets

### 4.2.1  Comparing calibration results against single reference datasets

In this section we analyse if HTESSEL configurations performing well against particular reference datasets also
perform well against other reference datasets, i.e. if a parameter set that yields for example good soil moisture performance also yields realistic LSTs. For this purpose we assess the performance of parameter perturbations performing best against particular reference datasets with respect to all other reference datasets in Figure 3. White colors mean that parameter perturbations which perform well against particular reference datasets (x-axis) also perform well against other reference datasets (y-axis). Vice versa, black colors indicate the opposite. Note
that in the case of a perfect model and perfect observations this plot would be all white. The many dark colored fields in Figure 3 indicate that different parameter perturbations perform best against different reference datasets. This can be explained by (1) equifinality (i.e. many different parameter sets leading to equally well performing model simulations) as there are 25 pre-selected well performing parameter sets among all 50 considered parameter sets, and by (2) inconsistencies within HTESSEL, especially between hydrological and
skin temperature-related processes. This is apparent as





for example HTESSEL configurations performing well in terms of LSTs yield particularly poor performance in terms of hydrology, and vice versa. These inconsistencies might be partly associated with missing processes in HTESSEL, for example the over-simplification that a single parameter represents the complex energy transfers between the top of the canopy and the underlying soil.

Investigating the role of equifinality we also perform this analysis with the 25 randomly chosen HTESSEL configurations only as displayed in Figure S2. In general, results are robust with respect to the employed set of parameter perturbations as indicated by the comparable patterns in the plots. This indicates a higher importance of inconsistent process representations in the HTESSEL model than of equifinality. Consequently, all the reference datasets considered in this study are needed to constrain HTESSEL, whereas for a perfect model, one

dataset would be sufficient.

An analysis of this kind can moreover be used to assess overall model performance which can be measured with the mean rank (grayness) across all tested combinations. Note, however, that the result is influenced by the selection and the quality of reference datasets.

### 4.2.2 Comparing calibration results against multiple reference datasets

In this section we assess the relative performance of the 50 HTESSEL configurations against multiple metrics, i.e. against several reference datasets using different measures of agreement between the reference and modelled data. In this context, we compute the ranks of all simulations against all considered metrics. Thereafter we calculate for each simulation the sum of the individual ranks obtained against the considered metrics. This sum of ranks is then a measure of overall performance of each simulation, and can be used to rank the overall

performance against multiple metrics.

In Figure 4, we test how the best- and worst-ranked parameter sets rank in the case that the validation metric(s) is/are replaced by the same number of other validation metric(s). For a perfect model and perfect observations we would find that the best- and worst-ranked parameter sets in terms of particular validation metrics are also best- and worst-ranked, respectively, when compared against other validation metrics. For HTESSEL we find

that for an increasing number of employed metrics, worst-performing parameter sets tend to also perform worse in the case of replaced validation metrics. This is a main result of this study, it means that poorly performing model configurations can be more robustly identified when assessing model performance against multiple validation metrics.

However, this behavior is not found for the best-performing parameter sets. There are two main reasons for this:

1. The poor correspondence of model performance against the considered single validation metrics as shown in Figure 1. The impact of the underlying partly inconsistent process representations within HTESSEL on the results in Figure 4 increases when using more validation metrics. When computing Figure 4 without the 2 validation metrics for which we find the highest average ranks in Figure 1 (i.e. in terms of which the performance of HTESSEL is most inconsistent with the performance against the

remaining metrics), the best-performing parameter sets are more robustly identified with increasing number of employed validation metrics (green dashed line). The opposite is found when re-computing figure 4 without the 2 validation metrics which correspond best with the remaining metrics, i.e. for which we find the lowest average ranks.




This implies that the better the model performance rankings against individual metrics correspond with each other (i.e. the more white color there is in Figure 1), the fewer metrics are required to robustly identify best- and worst-performing parameter sets. This supports the previously discussed importance of the mean rank in Figure 1, it furthermore provides an indication of the required number of metrics to calibrate the considered land surface model.

2.  Out of the 50 considered parameter sets, 25 were pre-selected as they performed particularly well. Hence the 50 parameter sets are therefore not randomly chosen but contain more well-performing configurations than expected by chance. Consequently, the performance of the best parameter sets are more similar than the performance of the worst-performing parameter sets such that for example ranks 1-5 might correspond to very similar performances. When computing the above analysis with only the 25 randomly selected parameter sets we find a more robust identification of well-performing parameter sets with increased validation metrics as shown in Figure S3, in contrast to the results in Figure 4.

### 4.2.3 Added value of using multiple reference datasets for coupled forecast skills

Adding to the above analyses we finally investigate if a more robust calibration of HTESSEL against multiple datasets yields more accurate weather forecasts. For this purpose we perform a similar analysis as in Figure 4. We test how the best- and worst-ranked parameter sets rank if the (uncoupled) HTESSEL validation metrics are replaced by (coupled) weather forecast skill metrics. The results are displayed in Figure 5.

Also in this analysis we find benefits of using multiple validation metrics. The parameter sets ranked best (worst) yield better (worse) forecast performance for an increasing number of employed validation metrics. It is another main finding of this study that land surface model calibration against multiple reference datasets instead of a single reference dataset can lead to better weather forecast performance. However, the differences between few and many considered metrics are smaller compared with Figure 4. This can be explained by (1) fewer tested parameter sets (lower signal-to-noise ratio) and by (2) low predictability of temperature and especially precipitation at long lead times (e.g. 31-45 days) such that there is poor forecast skill whatsoever independent of the HTESSEL configuration.

In summary, these results underline the importance of land surface (model calibration) for coupled weather forecast skill (Koster et al. 2011), especially in terms of anomaly correlations and bias, and to a weaker extent also in terms of reliability.

### 5. Conclusions

In this study we assess the performance of ECMWF's land surface model HTESSEL against comprehensive satellite-based land surface temperature observations. In this novel analysis, we focus on the mean LST bias and the simulated LST temporal dynamics, and find overall unsatisfactory performance. There is no region across Europe and Africa were both the mean LST and the dynamics are well captured by the model. The performance is poorest over high vegetation and improves for low or no vegetation. The particularly poor performance over high vegetation suggests models deficiencies related with the representation of the energy exchanges between the top of the canopy and the underlying soil.





Novel Earth observation data such as the LST dataset add to existing reference datasets, and we furthermore highlight the benefit of employing multiple reference datasets altogether in LSM analysis and calibration. They enable a more robust calibration and can therefore help to address the problem of constraining increasingly complex state-of-the-art LSMs. In this context, we also show that a better constrained LSM also contributes to improved weather forecasts.

Based on the analysis in Figure 4 we can even infer how many metrics (i.e. reference datasets and measures of agreement with it) are sufficient to robustly calibrate HTESSEL. While in the figure we only consider up to 5 metrics (as more can not be replaced in case of a total of 10 metrics), extrapolation of the results towards more metrics indicates that HTESSEL can be robustly calibrated against the 10 metrics used in this study. While this means that poorly performing parameter sets can be identified with the considered reference data, we can, however, not robustly determine best-performing parameter sets. This is due to shortcomings in physical process representations in HTESSEL where for instance the bias of the simulated daily LST range can not be improved without degrading the simulated hydrology.

While the above-described main results of this study should be very relevant for the land surface modelling community, there are caveats in our analysis:

1. The results are valid for the models used (HTESSEL and ECMWF ensemble forecasting system), and the parameters we chose to perturb. Future research is needed to analyze if the methodology and results are transferable to other models.

2. The results are based on the reference datasets and metrics applied here, and on their involved uncertainties. Even though we partly assessed the role of the suite of employed metrics (leaving out 2 metrics at a time), it is not clear if similar findings would be obtained with different reference datasets which inherit different uncertainty characteristics.

3. The assessment of the uncoupled HTESSEL simulations is partly based on the time period considered in the coupled forecasts (2001-2010). This might lead to an overestimation of the benefits of a robustly calibrated land surface model for coupled forecast performance.

4. Our findings might depend on the spatial (0.5°x0.5°, even though this had to be upscaled for the comparison with the ET and GRACE datasets) and temporal resolution (daily) used for the analysis.

Improved constraining of complex LSMs is essential to better exploit their potential and as a basis to represent additional physical processes or updated land-use maps as foreseen in future versions. A more robust model calibration probably also helps to improve the representation of quantities and processes which can not (yet) be constrained with existing observations (e.g. evapotranspiration, sensible heat flux). More physically-based model simulations can also foster improved understanding of (future) climate system functioning, which is particularly important in the context of climate change (IPCC, 2013) as the estimation of climate conditions outside the calibration range of a model is more reliable with more physically-based models. Finally, as shown in this study, a more robustly calibrated LSM also contributes to improved weather forecasts and is hence valuable for society.





**Acknowledgements**

We acknowledge the international soil moisture network (https://ismn.geo.tuwien.ac.at/, accessed on 13 September 2016) and the SwissSMEX network, http://www.iac.ethz.ch/group/land-climate-dynamics/research/swisssmex.html, accessed on 13 September 2016) for providing soil
moisture measurements, Sean Swenson and the NASA MEaSUREs Program for sharing GRACE land data (http://grace.jpl.nasa.gov, accessed on 13 September 2016), the LandFlux-EVAL ET dataset (http://www.iac.ethz.ch/group/land-climate-dynamics/research/landflux-eval.html, accessed on 13 September 2016), the European water archive in cooperation with the EU-FP6 project WATCH (http://www.eu-watch.org, accessed on 13 September 2016) for sharing streamflow
data, and the SEVIRI/MSG Land Surface Temperature data processed by the EUMETSAT LSA SAF (http://landsaf.ipma.pt, accessed on 13 September 2016) and re-distributed over a regular 0.05º×0.05º grid by the GlobTemperature portal (http://data.globtemperature.info/, accessed on 13 September 2016).

In addition, we acknowledge the E-OBS dataset established by the EU-FP6 project ENSEMBLES (http://ensembles-eu.metoffice.com, accessed on 13 September 2016) and the data providers in the ECA&D
project (http://www.ecad.eu, accessed on 13 September 2016) for sharing precipitation and temperature data. The research leading to these results has received funding from the European Union's Seventh Framework Programe (FP7) under the Grant Agreement 603608 (Earth2Observe). We thank Randy Koster for helpful discussions on the results.

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

Table 1: Overview of performed model experiments.

| Model | Type | Domain | Spatial resolution | Time period | Number of simulations |
|---|---|---|---|---|---|
| HTESSEL | uncoupled | Europe (10°W-50°E, 35°N-70°N) | 0.5°x0.5° | 1983-2014 | 50 different parameter sets |
| ECMWF ensemble prediction system | coupled forecasts | global | 0.7°x0.7° | 2001-2010 | 11 different parameter sets |





Table 2: Summary of perturbed model parameters and their characteristics (adapted from O16).

| Surface runoff effective depth | Skin conductivity | Minimum stomatal resistance | Maximum interception | Soil moisture stress function | Total Soil depth |
|---|---|---|---|---|---|
| Depth over which soil water content and soil water content at saturation are integrated vertically to derive maximum infiltration and eventually surface runoff | Determines coupling of surface energy balance with the underlying surface temperature; dependent on vegetation and stable/unstable conditions | Scales leaf area index in the computation of canopy resistance | Maximum water over a single layer of leaves or bare ground; used to define the interception tile fraction | Determines the shape (e.g. 1 for linear) of dependency of canopy resistance on soil moisture | Lower boundaries of the particular soil layers; top layer not impacted by perturbations to avoid impacts on the fast thermal response |





**Figure 1:** Evaluation of HTESSEL against LST data. We consider biases (top) and daily anomaly correlation (bottom) of mean daily LSTs and of the daily LST range. Left column displays results for all vegetation types, the other columns display results of particular vegetation types only. The dashed rectangle denotes the European area which most of this study is based on.





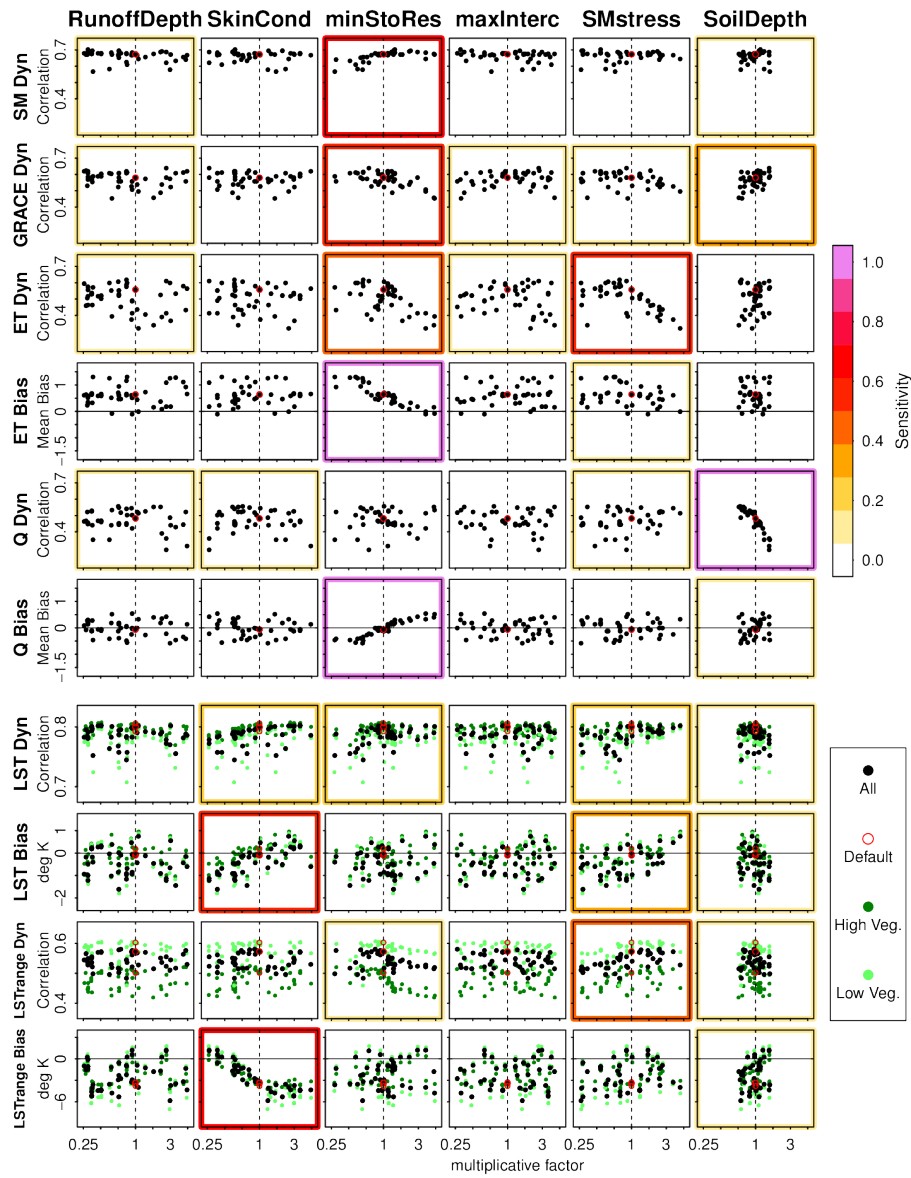

**Figure 2: Variations of of HTESSEL performance (over the European domain) in terms of anomaly correlations and biases against several reference datasets (y-axis) in response to variations in poorly constrained model parameters (x-axis). Upper part from O16. The color of each box indicates the sensitivity of the HTESSEL performance in terms of a particular metric against a particular parameter. Red circles indicate results for the default HTESSEL calibration, and green circles denote results for areas with low/high vegetation only.**





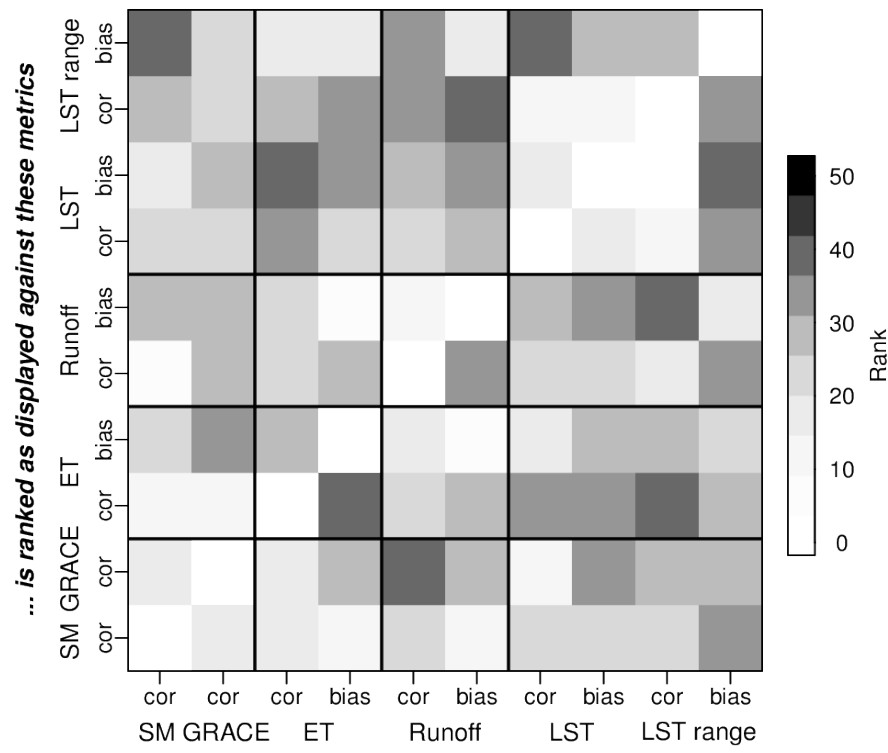

**Figure 3:** Performance of HTESSEL simulations with the different best-performing parameter sets as assessed against all particular metrics, respectively. For example the HTESSEL simulations with the parameter sets that yield best results in terms of ET bias (see corresponding column) also perform well in terms of runoff bias (white color) but not in terms of ET correlation (dark color).





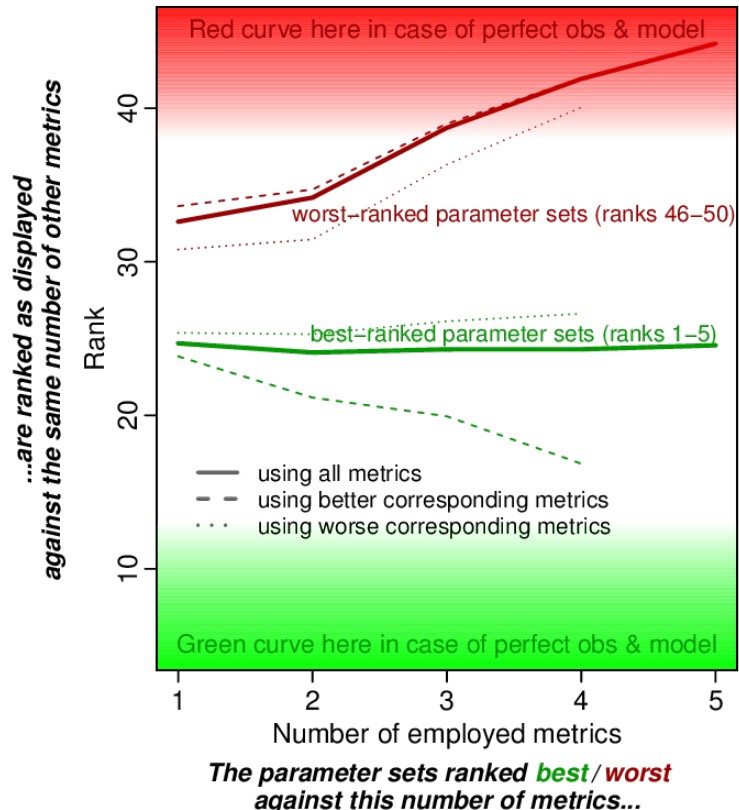

**Figure 4:** Comparing the rankings of HTESSEL simulations with all considered parameter sets when replacing a given set of evaluation metrics with an equal number of other metrics. The red curve displays the average rank of the previously worst-ranked parameter sets, and the green curve denotes the average rank of the previously best-ranked parameter sets. For each number of metrics, all possible combinations out of the 10 metrics employed in this study are considered and the mean results are displayed.

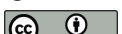


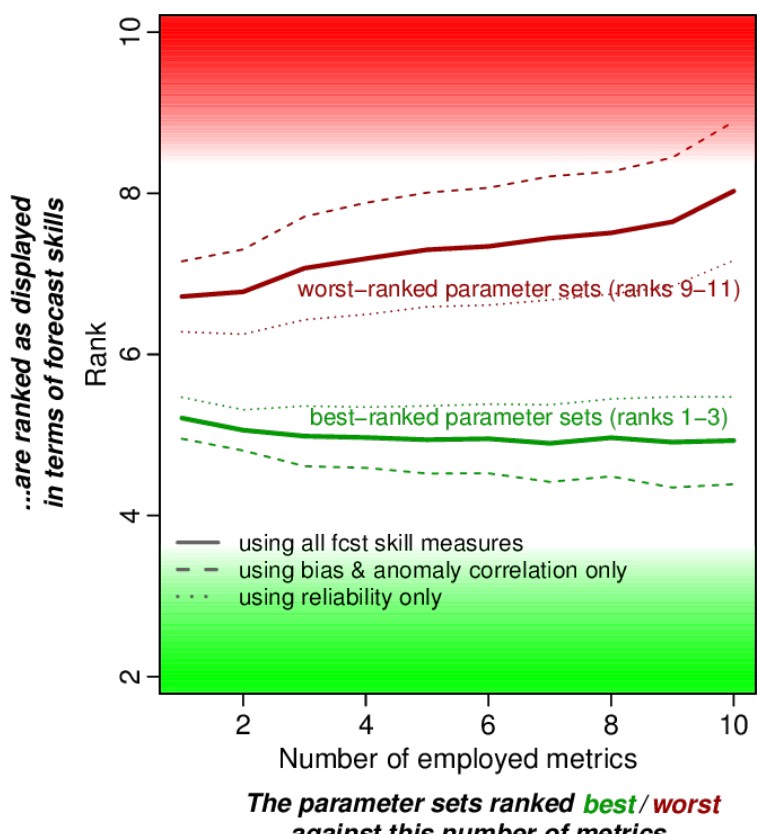

**Figure 5:** Similar to Figure 4 but comparing the performance (ranking) of HTESSEL simulations with 11 parameter sets across uncoupled evaluation (x axis) and coupled forecast skills (y axis).