# Peer review of "Advancing land surface model development with satellitebased Earth observations"

_Hydrology and Earth System Sciences, 2016_

## Referee Comment (RC1) · Anonymous Referee #1 · 20 Dec 2016

Review of paper: "Advancing land surface model development with satellite-based Earth observations"

By R. Orth et al.

General

This paper aims at validating the ECMWF surface model both coupled and uncoupled to the atmospheric model. The validation approach is rather complex, using several sources of satellite data. Ensemble technique is used with 6 parameters listed in Table 2 perturbed. The validation is both quantitative (for LST in Fig. 1) and qualitative to infer links between various parameters on the performance (Figs. 3-5). While the approach is interesting for the simultaneous examination of various causes of uncertainty, there is no clear path forward, notably to improve LST. It is commented that improving one LST may lead to deteriorating other aspects such as hydrology. It is implied that calibration of parameters will be best accomplished by simultaneously optimizing these. But how? The paper could be published after improving the flow of the arguments, and addressing points listed below.

Specific points

- Physical insight would be appreciated on LST results. Large range bias in southern Africa, for instance could be due to various factors. Which ones? Could it be for instance due to differences in surface emissivity used in retrievals versus that used in the model?
- Justify the need to run 45-day forecasts to infer the performance of a surface model. Seems quite long.
- The multiplicative factor s in section 2.2.3
- : 0.25-4.0 applied to selected parameters appear arbitrary. These should be based on assumed uncertainty.
- Similarly, why do you need a validation over many years to get Fig. 1? To do that, you presumably need initial ECMWF analysis which are based on different versions of the ECMWF model, which used different input data (notably satellite datasets).
- It seems that validation in coupled mode starts with an ECMWF analysis at each forecast, without actually cycling. Normally validation is done in an analysis cycle, which is quite demanding in terms of computer resources, but needed before implementing.
- Clarify the nature of the 25 parameter sets. How to do establish the "best parameter sets".

---

## Referee Comment (RC2) · Anonymous Referee #2 · 15 Feb 2017

(1) I prefer "help improve" to "help to improve" - Abstract Line 15 and everywhere

(2) Introduction Line 18: radiation at high - notice missing 't' in the manuscript

(3) Section 2.2.1: 1-year spin up is not adequate at all for offline simulations, usually we loop through 10-years at minimum to equilibrate initial states.

(4) a widely used term for the daily LST range is "Diurnal Temperature Range (DTR)", perhaps authors replace the term "the daily LST range" with DTR

(5) Section 3.1 Line 11 : acronym WFDEI - define it

(6) Section 3.1 (Forcing Data): claiming we used "observed meteorological" forcing may be an ambiguous statement. Perhaps you can rephrase the whole line "we used WFDEI bias corrected surface meteorological forcings "

[Figure]

(7) Section 3.2 A map showing Locations of soil moisture and streamflow stations will be helpful

(8) Line 21 the reader is referred to Table S1 in Orth and Seneviratne 2015 for details about soil moisture in-situ data, it would certainly helpful to have that table in this manuscript.

(9) Line 19: Data "are" available instead Data "is"

(10) Page 8, Line 15: "And while the ET data help to constrain HTESSSEL's energy balance during early years" - Sorry I did not get it, authors are merely using ET data to compute biases and skill in HTESSEL simulations, I believe it is not correct to claim HTESSEL's energy balance is constrained by ET data

(11) Figure 1: a) white-out the ocean b) what does color gray depict? c) colorbar is very poor, hard to figure out where the tick mark is d) 2 red and 2 blue levels look just the same e) you can say in the caption that the analysis used only northern hemisphere warm months May-Oct f) correlation coefficient or r2 - it does look like r2 (square of correlation) g) Instead DYN in Y-axis label used COR or R2 h) On authors discretion, you can have a 4 panel figure DTR BIAS/COR and LST mean BIAS/COR, because I did not see much value in breakdown analysis based on land cover type. i) also in correlation plots, white out the region where correlation is not statistically significant or (just random) for the confidence level of 95 % or something j) I assume Figure 1 is based on default HTSEESL parameter configuration. It would be interesting to see the same figure for the best set of parameters that your experiment yielded.

(12) Again colorbar 0.5 to 0.8 look just the same

---

## Referee Comment (RC3) · Anonymous Referee #3 · 28 Feb 2017

This manuscript by Rene Orth and fellow co-authors is a follow-on companion paper to the Orth et al. (2016) article, referred to in this manuscript as "O16". The new components in this HESS manuscript expanded over the O16 paper include evaluating their O16 simulations with geostationary satellite-based land surface temperature (LST) observations and exploring more the number of metrics and variables in combination with the LST observations. In the original O16 paper, the authors performed a parameter sensitivity analysis with 2300 offline HTESSEL simulations involving 6 in-situ observation sites (2 in Finland, 2 in Switzerland, and 2 in southern Italy). From these simulations, the authors pulled the best performing 25 (ranked) parameter datasets and then randomly selected another (different) 25 parameter datasets, along with the default parameter dataset. They identified in O16 six HTESSEL parameter types that the model may be most sensitive to and impacted by changes to. These same 6 are

used and evaluated in this study.

Major comments:

1) Please specify in this manuscript whether the exact 51 parameter datasets that were selected and used in the related offline HTESSEL simulations in O16 are the same as used in this study.

2) In section 2.3, are these performance measures applied the exact same way as in O16? The descriptions are different between the two papers. Also, is the calculated bias literally the difference between the average of the observations and average of the model simulation output, using the same overlapping years (e.g., 8 years of LST data)? Please describe further in your paper.

3) For the 11 soil moisture sites used in the evaluation, do they overlap at all with the original 6 sites used in the O16 calibration and evaluation paper? Or are these in addition to those 6 sites. Please specify and add in your figure 1 plot the location of those sites (or add a new figure highlighting the locations of the observations, including the 400+ streamflow gage sites).

4) In subsection 2.1.1, please further describe how the subgrid tiles of the HTESSEL model are organized (e.g., using top 5 dominant vegetation types per gridcell), or if just the dominant vegetation type tile is considered per gridcell.

5) In Figure 1, the authors consider grid cells where land cover is > 80%. However, most of the forested regions of "high vegetation" types look to be screened out from the analysis. Please explain if and how this might tie back to the LST screening process outlined in section 2.5 and how the vegetation tile process relates to it.

6) Figure 2 is essentially the same as Figure 3 in Orth et al. (2016) but with adding the LST reference dataset and focusing on just the 51 parameter datasets. In the description for this Figure 2, the authors however state in lines 13-15 on page 9 that the hydrology-based reference datasets are "sensitive mostly to stomatal conductivity".

Are you referring here to the minimum stomatal resistance parameter (minStoRes), which would match the results of Fig. 2, or the skin conductivity (SkinCond) parameter, which would not be right according the results shown? Similarly, what is stated for the skin temperature plots in Fig. 2 that the "performance is especially sensitive to minimal stomatal conductivity parameter". However, in Figure 2 the LST is highly sensitive to the "SkinCond" parameter. Please make the qualitative analysis description in these lines much more clearly reflect the results of Figure 2, and using the correct names of the parameters with the highest sensitivity.

7) In lines 35-37 on page 9, it is confusing when the authors state: "The many dark colored fields in Figure 3 indicate that different parameter perturbations perform best against different datasets.". However just above that statement, the authors say that the light colors indicate parameters that perform well against the reference datasets. Please rephrase these sentences to reduce any conflicting statements of what the light colors refer to.

8) The authors refer on line 2 of page 11, "i.e., the more white color there is in Figure 1)" . . . which areas of white color are you referring to? There are no white areas to refer to. Here you are referring to Figure 1 and on lines 2 and 4 of page 11. Please rewrite this paragraph to make it clearer and make sure you are referring to the correct figures(s).

9) One aspect of the geostationary satellite-based LST observations used as a reference dataset in this study that is hardly mentioned or discussed is what are the actual errors and uncertainty of this observed LST dataset itself. How does this factor in to the model evaluation (e.g., bias metric)? Please address this further in the paper. For example, if the observed LST dataset is heavily biased, if you were to calibrate the model to such biased observations, you could constrain the model to this bias.

10) In terms of the model spin-up period, 1-year spinup is not usually sufficient, especially for soil moisture and temperature in desert regions, where it is has been shown

that it can take more than 20 years for such locations to reach equilibrium. Please consider running a longer spinup period and rerun the analysis for the LST evaluation. Would the longer spinup affect the model LST bias in the bare soil regions, like Sahara desert.

Minor Comments:

1) In subsection 3.2.2., please provide additional details on the "E-OBS" dataset, which is supposedly used in validating the coupled forecasts. 2) On page 4, line 5, need to place a period after "…Balsamo et al., 2011)" reference.

3) In Figure 1, please define in the figure caption and in the main text what "DYN" stands for (which is only found on the "y-axis" labels for the bottom two rows of plots).

4) Also regarding the Figure 1 description in Section 4, please confirm in the text the "direction of bias", e.g., obs – model (default parameters, based on the bias definition in section 2.3). This is to help clarify when explaining in the two top rows of Figure 1 that the model is greatly underestimating the diurnal range in the more vegetated regions and overestimating in desert regions, like the Sahara.

5) What does the gray areas indicate in Figure 1?

6) On page 10, lines 32-36, this appears to be a run-on sentence. Please consider making this sentence into two with the start of the new sentence begin on line 30, "The best-performing parameter sets …".

7) Also for the point #1 starting on line 30, it is difficult to follow. Please try to more concisely explain your results and point made here relative to Figure 4. Which "two validation metrics" are you referring to?

8) On page 11, line 33, make "were" to "where", as "where both the mean LST …".

9) For the LST data background, is what is described in section 2.5 the same dataset as describd in section 3.2.1 on page 8? If so, please consider combining into one of

the given sections.

10) For the Section 3.1, please add more to the section on the WFDEI forcing dataset.

References: Orth, R., E. Dutra, and F. Pappenberger (2016), Improving weather predictability by including land-surface model parameter uncertainty. Mon. Weather Rev., 144(4), 1551-1569

―――――――――――――――――

---

## Author Comment (AC1) · 28 Mar 2017

We thank all reviewers for their valuable feedback which helped us to improve the manuscript. In response, aside from several minor corrections and clarifications, we have introduced the following main changes to the manuscript:

(1) Longer spin-up period in HTESSEL simulations. It has been increased from 1 year to 6 years. The simulations start in 1983, and validation of these simulations against observations starts in 1989. This is accomplished by starting the runoff validation only in 1989 instead of 1984. Validation in terms of other considered variables is not affected as the corresponding reference datasets only start in 1989 or later. All Figures have been re-computed taking into account the updated runoff validation. The changes compared with the previous version of the manuscript are minor.

(2) Comprehensive revision Figure 1. Results from specific vegetation types are now displayed as barplots. Furthermore, several labels and the colorbars have been updated, and the statistical significance of the displayed correlations is now shown.

(3) Additional figure displaying locations of soil moisture and runoff measurement stations from which data is employed in this study.

**Reviewer #1**

General

This paper aims at validating the ECMWF surface model both coupled and uncoupled to the atmospheric model. The validation approach is rather complex, using several sources of satellite data. Ensemble technique is used with 6 parameters listed in Table 2 perturbed. The validation is both quantitative (for LST in Fig. 1) and qualitative to infer links between various parameters on the performance (Figs. 3-5). While the approach is interesting for the simultaneous examination of various causes of uncertainty, there is no clear path forward, notably to improve LST.

A1: The goal of our study is to highlight the opportunities arising from the validation of a land surface model against novel Earth observation datasets, and to show the benefits of employing them alongside existing reference datasets. This includes the illustration of identifying formerly unknown shortcomings, as in the case of LST. It is, however, beyond the scope of the present study to address these shortcomings. Nevertheless our analyses form a valuable basis for future development efforts.
This clarification has been added to the conclusions section.

It is commented that improving one LST may lead to deteriorating other aspects such as hydrology. It is implied that calibration of parameters will be best accomplished by simultaneously optimizing these. But how?

A2: We thank the reviewer for this comment. Model optimization against multiple reference datasets using various metrics of agreement is indeed a major recommendation arising from this study. This can be best achieved by using more comprehensive objective functions when calibrating models. These functions should be composed of various parts, each of which assesses the agreement between the model simulations and a particular reference dataset using a particular metric of agreement. This way, model parameters can be adjusted more reliably to yield reasonable model performance in terms of various variables, and to

capture possible couplings between them.
This argumentation has been added to the conclusions section.

The paper could be published after improving the flow of the arguments, and addressing points listed below.
Specific points
- Physical insight would be appreciated on LST results. Large range bias in southern Africa, for instance could be due to various factors. Which ones? Could it be for instance due to differences in surface emissivity used in retrievals versus that used in the model?

A3: The differences between model skin temperature and LST estimates may arise from both model or LST uncertainties. The LST data used in this study (based on LSA SAF estimates from SEVIRI/MSG) have been thoroughly validated over the last 10-years against in situ observations and also via comparisons with other satellite retrievals (e.g., Trigo et al, 2008; Ermida et al., 2014; Göttsche et al., 2016). The validation with ground data indicates that under most conditions, LST biases are below 1K; these can however reach up to 2.5K in the case of a very moist atmosphere.
This point has been added to section 3.2.1.
Although there are several factors conditioning the uncertainty in LST satellite estimates (including total column water vapour, view angle and surface emissivity), the differences between model and satellite LST are often beyond the expected accuracy of the latter.
This point has been added to section 4.1.
Regarding the comment on the emissivity differences between model and those used in the retrievals, it should be noted that these are not actually the same variable: a broad-band (whole infrared range 3-100 micro-m) emissivity is used in the model to determine the total long-wave flux leaving the surface (reflected and emitted), while the LST satellite estimates make use of land surface emissivities within the range of the used channels (around 10.8 micro-m and 12.0 micro-m in the case of SEVIRI/MSG LST).

- Justify the need to run 45-day forecasts to infer the performance of a surface model. Seems quite long.

A4: We think that the computational costs of forecasts with lead times of 45 days is justified as the land surface can potentially influence forecast skills at such long lead times thanks to its profound persistence characteristics. And indeed, the relationship illustrated in Figure 5 is found at all considered lead times.
These information have been added to sections 2.2.2 (first point) and 4.2.3 (latter point).

- The multiplicative factor s in section 2.2.3
- :        0.25-4.0 applied to selected parameters appear arbitrary. These should be based on assumed uncertainty.

A5: The multiplicative factors were selected to cover a range of possible values. This is rather arbitrary as the reviewer pointed out. However, although the parameters are not directly observable, they are associated with physical processes, and values beyond the selected ranges would not be acceptable. As illustrated in Figure 2, the selected range is adequate for our purpose, namely to study the sensitivity of model performance to particular

parameter values. Note also that using a different range of e.g. half the actual size (0.5-2) would not affect the results shown in Figure 2.
This point has been clarified in section 2.2.3 of the manuscript.

- Similarly, why do you need a validation over many years to get Fig. 1? To do that, you presumably need initial ECMWF analysis which are based on different versions of the ECMWF model, which used different input data (notably satellite datasets).

A6: Several years were included in the validation to cover as much as possible inter-annual variability, and to prevent an influence of large-scale climate anomalies (e.g. El-Nino) on our results. By using the ECMWF ERA-Interim reanalysis to force HTESSEL, the forcing model version is the same throughout the period. As the reviewer points out, there are changes in the input data used by the data assimilation during that period. However, for the period 2007-2014 those changes are minor compared with changes in previous decades.

- It seems that validation in coupled mode starts with an ECMWF analysis at each forecast, without actually cycling. Normally validation is done in an analysis cycle, which is quite demanding in terms of computer resources, but needed before implementing.

A7: This is a very important point raised by the reviewer. We could not afford the computational cost of a long-term reanalysis spanning all the initialization period. However, we replicate what is currently done to initialize the ECMWF operational hindcasts of the sub-seasonal and seasonal systems, that are initialized from ERA-Interim for the atmosphere and from a land-surface only simulation using exactly the same land surface model version as in the forecasts. This is done replicating the ERA-Interim/land dataset but with the land-surface model equal to the one used in the forecast (i.e. with the set of perturbed parameters).
This point has been clarified in section 2.2.2 of the manuscript.

- Clarify the nature of the 25 parameter sets. How to do establish the "best parameter sets".

A8: In O16, all considered >2000 parameter sets were ranked in terms of each reference dataset, and then for each particular parameter set the sum of all ranks was computed. In this context, O16 used reference data of soil moisture, terrestrial water storage, runoff, and evapotranspiration.
This point has been clarified in section 2.2.3 of the manuscript.

**Reviewer #2**

(1) I prefer "help improve" to "help to improve" - Abstract Line 15 and everywhere

B1: Corrected.

(2) Introduction Line 18: radiation at high - notice missing 't' in the manuscript

B2: Corrected.

(3) Section 2.2.1: 1-year spin up is not adequate at all for offline simulations, usually we loop through 10-years at minimum to equilibrate initial states.

B3: We thank the reviewer for this comment. As explained in point (1) in the introduction above, we have extended the spin-up period to 6 years. This is accomplished by starting the runoff validation only in 1989 instead of 1984. Validation in terms of other considered variables is not affected as the corresponding reference datasets only start in 1989 or later. All Figures have been re-computed taking into account the updated runoff validation. The changes compared with the previous version of the manuscript are minor.

(4) a widely used term for the daily LST range is "Diurnal Temperature Range (DTR)", perhaps authors replace the term "the daily LST range" with DTR

B4: A comment on this point has been added to section 2.5 of the manuscript.

(5) Section 3.1 Line 11 : acronym WFDEI - define it

B5: Done.

(6)  Section 3.1 (Forcing Data): claiming we used "observed meteorological" forcing may be an ambiguous statement. Perhaps you can rephrase the whole line "we used WFDEI bias corrected surface meteorological forcings "

B6: We corrected "observed" to "observation-based".

(7) Section 3.2 A map showing Locations of soil moisture and streamflow stations will be helpful

B7: We thank the reviewer for this comment. The map has been added as Figure S1.

(8) Line 21 the reader is referred to Table S1 in Orth and Seneviratne 2015 for details about soil moisture in-situ data, it would certainly helpful to have that table in this manuscript.

B8: We have included the Table in the manuscript as Table S1.

(9) Line 19: Data "are" available instead Data "is"

B9: Corrected.

(10) Page 8, Line 15: "And while the ET data help to constrain HTESSSEL's energy balance during early years" - Sorry I did not get it, authors are merely using ET data to compute biases and skill in HTESSEL simulations, I believe it is not correct to claim HTESSEL's energy balance is constrained by ET data

B10: We agree with the reviewer and corrected it.

(11) Figure 1: a) white-out the ocean b) what does color gray depict? c) colorbar is very poor, hard to figure out where the tick mark is d) 2 red and 2 blue levels look just the same e) you can say in the caption that the analysis used only northern hemisphere warm months May-Oct f) correlation coefficient or r2 - it does look like r2 (square of correlation) g) Instead DYN in Y-axis label used COR or R2 h) On authors discretion, you can have a 4 panel figure DTR BIAS/COR and LST mean BIAS/COR, because I did not see much value in breakdown analysis based on land cover type. i) also in correlation plots, white out the region where correlation is not statistically significant or (just random) for the confidence level of 95 % or something j) I assume Figure 1 is based on default HTSEESL parameter configuration. It would be interesting to see the same figure for the best set of parameters that your experiment yielded.

B11: Implementing the reviewers suggestions, we have comprehensively revised Figure 1.

(12) Again colorbar 0.5 to 0.8 look just the same

B12: The color bar in Figure 2 was improved accordingly.

**Reviewer #3**

This manuscript by Rene Orth and fellow co-authors is a follow-on companion paper to the Orth et al. (2016) article, referred to in this manuscript as "O16". The new components in this HESS manuscript expanded over the O16 paper include evaluating their O16 simulations with geostationary satellite-based land surface temperature (LST) observations and exploring more the number of metrics and variables in combination with the LST observations. In the original O16 paper, the authors performed a parameter sensitivity analysis with 2300 offline HTESSEL simulations involving 6 in-situ observation sites (2 in Finland, 2 in Switzerland, and 2 in southern Italy). From these simulations, the authors pulled the best performing 25 (ranked) parameter datasets and then randomly selected another (different) 25 parameter datasets, along with the default parameter dataset. They identified in O16 six HTESSEL parameter types that the model may be most sensitive to and impacted by changes to. These same 6 are used and evaluated in this study.
Major comments:
1) Please specify in this manuscript whether the exact 51 parameter datasets that were selected and used in the related offline HTESSEL simulations in O16 are the same as used in this study.

C1: Yes, the same 50 parameter sets from O16 are used in this study. This has been clarified in the methods section of the manuscript.

2) In section 2.3, are these performance measures applied the exact same way as in O16? The descriptions are different between the two papers. Also, is the calculated bias literally

the difference between the average of the observations and average of the model simulation output, using the same overlapping years (e.g., 8 years of LST data)? Please describe further in your paper.

C2: The measures have been applied previously in O16. And the biases are computed using the exact same time periods for the reference and model data. Both points have been clarified in section 2.3.

3) For the 11 soil moisture sites used in the evaluation, do they overlap at all with the original 6 sites used in the O16 calibration and evaluation paper? Or are these in addition to those 6 sites. Please specify and add in your figure 1 plot the location of those sites (or add a new figure highlighting the locations of the observations, including the 400+ streamflow gage sites).

C3: The 6 soil moisture measurement sites used in O16 are all contained in the set of 11 soil moisture stations used for model validation. The new Figure S1 illustrates the locations of these sites, and the length of the available data record from each site.

4) In subsection 2.1.1, please further describe how the subgrid tiles of the HTESSEL model are organized (e.g., using top 5 dominant vegetation types per gridcell), or if just the dominant vegetation type tile is considered per gridcell.

C4: In HTESSEL there are two up to 9 tiles, with two for vegetation: high and low, and on these only the dominant is represented. For example, we can have a gridcell with 30% evergreen broadleaf trees (for the high vegetation) and 40% short grass (for the low vegetation), and the remaining with a lake, but only 1 type of high and 1 type of low vegetation is allowed.
This information has been added to section 2.1.1.

5) In Figure 1, the authors consider grid cells where land cover is > 80%. However, most of the forested regions of "high vegetation" types look to be screened out from the analysis. Please explain if and how this might tie back to the LST screening process outlined in section 2.5 and how the vegetation tile process relates to it.

C5: The cloud cover is often high over grid points where "High vegetation" type dominates (e.g., tropical forests, boreal forests), which eventually leads to a significant screening of that land cover. Hence, no LST reference data is available e.g. over tropical Africa to validate HTESSEL's skin temperature in Figure 1. The vegetation tile process does not relate to this. This point has been clarified in section 4.1.

6) Figure 2 is essentially the same as Figure 3 in Orth et al. (2016) but with adding the LST reference dataset and focusing on just the 51 parameter datasets. In the description for this Figure 2, the authors however state in lines 13-15 on page 9 that the hydrology-based reference datasets are "sensitive mostly to stomatal conductivity".
Are you referring here to the minimum stomatal resistance parameter (minStoRes), which would match the results of Fig. 2, or the skin conductivity (SkinCond) parameter, which would not be right according the results shown?

C6: We thank the reviewer for pointing us to this mistake. It should be the stomatal resistance. We corrected this.
Note furthermore that Figure 2 is not actually the same as Figure 3 in O16. That figure showed model performance results for the selected 6 sites, whereas Figure 2 in this study shows model performance results averaged across Europe. We clarified this point in the figure caption.

Similarly, what is stated for the skin temperature plots in Fig. 2 that the "performance is especially sensitive to minimal stomatal conductivity parameter". However, in Figure 2 the LST is highly sensitive to the "SkinCond" parameter. Please make the qualitative analysis description in these lines much more clearly reflect the results of Figure 2, and using the correct names of the parameters with the highest sensitivity.

C7: Also this mistake has been corrected.

7) In lines 35-37 on page 9, it is confusing when the authors state: "The many dark colored fields in Figure 3 indicate that different parameter perturbations perform best against different datasets.". However just above that statement, the authors say that the light colors indicate parameters that perform well against the reference datasets. Please rephrase these sentences to reduce any conflicting statements of what the light colors refer to.

C8: We rephrased this paragraph to clarify that dark colors indicate that the parameter perturbations performing best against particular reference datasets are different, i.e. there is no parameter perturbation that performs best in all respects.

8) The authors refer on line 2 of page 11, "i.e., the more white color there is in Figure 1)" ... which areas of white color are you referring to? There are no white areas to refer to. Here you are referring to Figure 1 and on lines 2 and 4 of page 11. Please rewrite this paragraph to make it clearer and make sure you are referring to the correct figures(s).

C9: We thank the reviewer for noting this. We refer to Figure 3 and corrected the paragraph accordingly.

9) One aspect of the geostationary satellite-based LST observations used as a reference dataset in this study that is hardly mentioned or discussed is what are the actual errors and uncertainty of this observed LST dataset itself. How does this factor in to the model evaluation (e.g., bias metric)? Please address this further in the paper. For example, if the observed LST dataset is heavily biased, if you were to calibrate the model to such biased observations, you could constrain the model to this bias.

C10: The differences between model skin temperature and LST estimates may arise from both model or LST uncertainties. The LST data used in this study (based on LSA SAF estimates from SEVIRI/MSG) have been thoroughly validated over the last 10-years against in situ observations and also via comparisons with other satellite retrievals (e.g., Trigo et al, 2008; Ermida et al., 2014; Göttsche et al., 2016). The validation with ground data indicates that under most conditions, LST biases are below 1K; these can however reach up to 2.5K in the case of a very moist atmosphere.
This point has been added to section 3.2.1.

Although there are several factors conditioning the uncertainty in LST satellite estimates (including total column water vapour, view angle and surface emissivity), the differences between model and satellite LST are often beyond the expected accuracy of the latter. This point has been added to section 4.1.

10) In terms of the model spin-up period, 1-year spinup is not usually sufficient, especially for soil moisture and temperature in desert regions, where it is has been shown that it can take more than 20 years for such locations to reach equilibrium. Please consider running a longer spinup period and rerun the analysis for the LST evaluation. Would the longer spinup affect the model LST bias in the bare soil regions, like Sahara desert.

C11: We thank the reviewer for this comment. As explained in point (1) in the introduction above, we have extended the spin-up period to 6 years. This is accomplished by starting the runoff validation only in 1989 instead of 1984. Validation in terms of other considered variables is not affected as the corresponding reference datasets only start in 1989 or later. All Figures have been re-computed taking into account the updated runoff validation. The changes compared with the previous version of the manuscript are minor.

Minor Comments:
1) In subsection 3.2.2., please provide additional details on the "E-OBS" dataset, which is supposedly used in validating the coupled forecasts.

C12: More detail on the E-OBS dataset has been included in section 3.2.2.

2) On page 4, line 5, need to place a period after ". . .Balsamo et al., 2011)" reference.

C13: Corrected.

3) In Figure 1, please define in the figure caption and in the main text what "DYN" stands for (which is only found on the "y-axis" labels for the bottom two rows of plots).

C14: We changed "DYN" to "COR" to clarify that we are referring to anomaly correlations.

4) Also regarding the Figure 1 description in Section 4, please confirm in the text the "direction of bias", e.g., obs – model (default parameters, based on the bias definition in section 2.3). This is to help clarify when explaining in the two top rows of Figure 1 that the model is greatly underestimating the diurnal range in the more vegetated regions and overestimating in desert regions, like the Sahara.

C15: Actually we compute the bias as model - obs. We corrected this point in section 2.3, and furthermore clarify it in Figure 1, and in the corresponding text.

5) What does the gray areas indicate in Figure 1?

C16: These are ocean areas, and areas where not LST data is available to validate HTESSEL's skin temperature. This is clarified in the figure caption.

6) On page 10, lines 32-36, this appears to be a run-on sentence. Please consider making this sentence into two with the start of the new sentence begin on line 30, "The best-performing parameter sets . . .".

C17: We shortened this sentence.

7) Also for the point #1 starting on line 30, it is difficult to follow. Please try to more concisely explain your results and point made here relative to Figure 4. Which "two validation metrics" are you referring to?

C18: We rephrased this paragraph, and specified which metrics we are referring to.

8) On page 11, line 33, make "were" to "where", as "where both the mean LST . . .".

C19: Corrected.

9) For the LST data background, is what is described in section 2.5 the same dataset as describd in section 3.2.1 on page 8? If so, please consider combining into one of the given sections.

C20: We decided to keep the two paragraphs in sections 2.5 and 3.2.1. Section 2.5 described the methodology we use to process the raw LST data, whereas in section 3.2.1 the actual product is introduced, including corresponding references. However, in section 2.5 we now point the reader to section 3.2.1, and vice versa.

10) For the Section 3.1, please add more to the section on the WFDEI forcing dataset.

C21: More information on this dataset has been added to section 3.1.

---

## Author Response (AR2)

Dear Authors,

Thank you for uploading the revised version of this manuscript. There are still some remaining minor comments.

Best,
Shrad

We are thankful to the Editor for these comments. We have addressed the recommended minor changes as detailed below.

Section 3.1: Please change "employ the WFDEI dataset (Watch Forcing Data methodology applied to Era-Interim data, Weedon et al. 2014)" to Watch Forcing Data methodology applied to Era-Interim (WFDEI, Weedon et al. 2014) dataset.

A1: Done.

In response to Reviewer #2, comment #7, I do not see a map showing soil moisture and streamflow stations in the manuscript. Likewise, I also do not see the Table S1. Perhaps you have uploaded an older version of the supplementary material?

A2: We will make sure to upload the updated supplementary material which includes the map and Table S1.

In response to Reviewer #3, comment #10 on section 3.1. What additional information regarding the forcing datasets has the authors included that was not present in the previous version. As far as I can tell the only difference in this version with the previous version is that this version contains the full form of WFDEI.

A3: We have added more detail on the WFDEI dataset by mentioning that it is based on ERA-Interim data, and additionally adjusted with respect to monthly gridded observations of temperature, precipitation, cloud cover, and atmospheric aerosol loading.

Page 1: Please change "difficulties to correctly capture" to "difficulties in correctly capturing".

A4: Done.

Page 2: Line 6: Please change "This lead" to "This leads".

A5: Done.

Page 2, Line 9: Please change "which become" to "which have become".

A6: Done.

Page 2: Line 33: Please change "helps to better" to "help better".

A7: Done.

Page 2, Line 37: Please change "investigate the role of the land surface model calibration for the skill of" to "investigate the influence of the land surface model calibration on the skill of".

A8: Done.

Page 3 Line 8: Please change "skills" to "skill".

A9: Done.

Page 4, Line 26: Not sure why "sub" needs to be within parenthesis here?

A10: We have removed the brackets.

Page 7: Line 21. I do not see the map in Figure S1.

A11: See answer A2.

4.2.3. Please change "skills" to "skill".

A12: Done.

[revised manuscript text omitted]